# Chitosan-G-Glycidyl Methacrylate/Au Nanocomposites Promote Accelerated Skin Wound Healing

**DOI:** 10.3390/pharmaceutics14091855

**Published:** 2022-09-02

**Authors:** Héctor A. López-Muñoz, Mauricio Lopez-Romero, Moises A. Franco-Molina, Alejandro Manzano-Ramirez, Cristina Velasquillo, Beatriz Liliana España-Sanchez, Ana Laura Martinez-Hernandez, Hayde Vergara-Castañeda, Astrid Giraldo-Betancur, Sarai Favela, Rogelio Rodriguez-Rodriguez, Juan Carlos Mixteco, Juan Carlos Tapia-Picazo, Diana G. Zarate-Triviño, Evgeny Prokhorov, Gabriel Luna-Barcenas

**Affiliations:** 1Nanosciences Program, Cinvestav, Ciudad de Mexico 07360, CDMX, Mexico; 2Cinvestav Queretaro, Queretaro 76230, QE, Mexico; 3Immunology and Virology Department, Biological Sciences Faculty, Universidad Autonoma de Nuevo León, San Nicolás de los Garza 66450, NL, Mexico; 4Instituto Nacional de Rehabilitacion, Ciudad de Mexico 14389, CDMX, Mexico; 5CONACYT—CIDETEQ, SC, Pedro Escobedo 76703, QE, Mexico; 6Tecnologico Nacional de Mexito-Instituto Tecnologico de Queretaro, Queretaro 76000, QE, Mexico; 7Facultad de Medicina, Universidad Autonoma de Queretaro, Queretaro 76176, QE, Mexico; 8CONACYT—Cinvestav Queretaro, Queretaro 76230, QRO, Mexico; 9Instituto de Ingenieria y Tecnologia, Universidad Autonoma de Ciudad Juarez, Ciudad Juarez 38584, CHIH, Mexico; 10Centro Universitario de los Valles, Universidad de Guadalajara, Ameca 46600, JA, Mexico; 11Tecnologico Nacional de Mexico-Instituto Tecnologico de Aguascalientes, Aguascalientes 20256, AG, Mexico

**Keywords:** Au-chitosan nanocomposite, gold nanoparticles, skin wound

## Abstract

Herein, we report the synthesis of Au nanoparticles (AuNPs) in chitosan (CTS) solution by chemically reducing HAuCl_4_. CTS was further functionalized with glycidyl methacrylate (chitosan-g-glycidyl methacrylate/AuNP, CTS-g-GMA/AuNP) to improve the mechanical properties for cellular regeneration requirements of CTS-g-GMA/AuNP. Our nanocomposites promote excellent cellular viability and have a positive effect on cytokine regulation in the inflammatory and anti-inflammatory response of skin cells. After 40 days of nanocomposite exposure to a skin wound, we showed that our films have a greater skin wound healing capacity than a commercial film (TheraForm^®^), and the presence of the collagen allows better cosmetic ave aspects in skin regeneration in comparison with a nanocomposite with an absence of this protein. Electrical percolation phenomena in such nanocomposites were used as guiding tools for the best nanocomposite performance. Our results suggest that chitosan-based Au nanocomposites show great potential for skin wound repair.

## 1. Introduction

The synthesis of gold nanoparticles for nanocomposite formulation has received considerable attention in the last decade. Several reports describe synthesis processes in which the produced nanoparticles have homogeneous sizes and shapes [1,2,3]. Gold nanoparticles (AuNPs) have attracted substantial attention due to their optical, electrical, mechanical, and biomedical properties [4], which make them potential candidates for applications in various areas such as drug delivery, development of sensors, catalysis, and environmental remediation [5,6]. Animal models, such as mice, are often used in wound healing research, and it has been observed that low concentrations of AuNPs increase cellular proliferation and the production of cytokines [7,8]. The nanosized Au particles offer the possibility of attaching molecules to their surface; these particles can interact with membrane lipids and can penetrate the stratum corneum of the skin [9]. Au nanocomposites have been used as anti-inflammatory agents because they can inhibit the NF-κB transcription factor. Additionally, it has been shown that AuNPs can reduce reactive oxygen species and that a complex of vitamin E-gold nanoparticles can improve the antioxidant activity acting as a catalyst in nicotinamide adenine dinucleotide hydrogen (NADH) to nicotinamide adenine dinucleotide (NAD) complex reactions [10].

Chitosan (CTS) is a natural-origin polymer that is biocompatible, biodegradable, and non-toxic, with antibacterial properties due to its chemical structure; a surface-active film can be generated that can interact with different organic molecules such as amino acids, lipids, nucleic acids, and proteins. Muzzarelli et al. (2009) demonstrated that CTS induces macrophage activation, re-epithelialization, and cicatrization. Other studies determined that CTS increases phagocytosis, TGF-β, platelets growth factor production, IL-8, and fibroblast proliferation, and decreases IL-1 production; all these properties induce granular tissue and regulate collagen fiber deposition, which can contribute to the wound healing process. Some nanoparticles with CTS have been reported for tissue regeneration or engineering. For example, CTS with silver nanoparticles showed capacity as a wound dressing due to its significant antibacterial properties and cell biocompatibility with human fibroblast.

On the other hand, CTS with BaTiO_3_ nanostructure showed excellent biocompatibility with the same kind of cells [11]. However, a general problem with CTS is its lack of mechanical properties, making it challenging to use these biomaterials for other cells. Therefore, the functionalization of the CTS with other types of synthetic and natural polymers is necessary to improve these properties.

Glycidyl methacrylate (GMA) is a synthetic monomer that can be functionalized with CTS in an aqueous solution; this produces two reactive functional groups that can react with amines, carboxylic acids, and hydroxides group to produce a hybrid natural–synthetic hydrogel (CTS-g-GMA), with an improvement of mechanical properties up to 25% compared to pristine CTS [12]. Our research group reported the first use of this biomaterial for cellular regeneration; we used the CTS-g-GMA to regenerate bone, spinal cord, and cartilage with promising results [13,14,15]. However, we want to improve the healing and regeneration process by including proteins that are naturally expressed during cellular repair; this change could improve the process and create a better or more natural healing or regenerative process. For instance, collagen is a protein mainly found in the extracellular matrix component of several tissues such as bone, tendons, and skin [16]. Type I collagen is present in human skin at 75–80% [17].

This study aims to synthesize, characterize, and assess the cellular viability and healing process in a rat model of different nanocomposites based upon GMA-functionalized chitosan and Au nanoparticles (CTS-g-GMA-AuNPs), and GMA-functionalized chitosan, Au nanoparticles, and type I collagen (col) ((CTS-g-GMA)-AuNPs-col). In addition, we attempt to correlate the structural properties of such nanocomposites with their efficacy in promoting a healing process; we also study the spectroscopic fingerprints and electrical percolation behaviors of such nanocomposites.

## 2. Materials and Methods

### 2.1. Nanocomposites Preparation

Reagents from Sigma Aldrich^®^ were used as received and without further purification. First, the gold nanocomposites (CTS-AuNPs) were synthesized by chemical reduction, dissolving 1 mL of 2% *w*/*v* of CTS (with medium molecular mass 300,000 g/mol and 85% degree of deacetylation) in 0.4 M acetic acid solution; afterward, 3 mL of 0.15, 0.3, 0.6, 1.07 mM HAuCl_4_ solutions were added to the mixture. The solutions were mixed and heated at 90 °C under magnetic stirring until the solution changed color from slightly yellow to red wine.

This material was synthesized at the stoichiometric mass ratio of CTS:GMA at 1:4; 0.25 g of CTS was dissolved in acetic acid solution at 0.4 M and 1 g of GMA was added to this solution. The solution was kept at 60 °C for 2 h at constant magnetic stir and nitrogen flow; precipitation was induced with acetonitrile according to the methodology reported by Elizalde et al. (2013) [12].

To elaborate the (CTS-g-GMA)-AuNPs nanocomposite, CTS-g-GMA was dispersed in distilled water for 24 h. Then, the CTS-HAuCl_4_ solutions with different concentrations of HAuCl_4_ were added, and the solution was kept at 60 °C for one hour until a homogeneous solution was obtained. The CTS is a reduction and stabilizing agent [18,19,20,21,22].

On the other hand, a third nanocomposite (CTS-g-GMA)-AuNPs-Col was prepared by using the procedure described above with one additional step at the end of the process: the incorporation of 0.3% of solution of type I collagen in acetic acid under constant magnetic stirring for 2 h at room temperature.

Thin films with different composites (thicknesses varied from 0.3 mm to a few millimeters for the different spectroscopic and in vivo studies) were prepared by the solvent cast method, i.e., pouring the final solution into a plastic Petri dish; then, they were placed in an oven at 60 °C for 24 h to allow evaporation of the solvent.

### 2.2. Characterization Studies

All materials were characterized by infrared spectroscopy (FTIR). The spectra between 4000 and 400 cm^−1^ were obtained using an FTIR spectrophotometer (Perkin Elmer Spectrum 1 Model). All spectra were recorded with a resolution of 4 cm^−1^ and 16 cm^−1^ times scanning using the transmission technique. Only one material for the HAuCl_4_ concentration was measured.

The film morphology of (CTS-g-GMA)-AuNPs and CTS/AuNPs was analyzed using a JEOM JSM-7401F field emission scanning electron microscope.

Crystal structure analysis was performed using a Rigaku diffractometer ULTIMA IV, equipped with CuKα radiation (λ = 1.5406 Å).

The ultraviolet-visible spectrum (UV–vis spectrometer Agilent 8453) was used to determine the sizes of the Au nanoparticles by detecting the maximum absorption band in the visible region. The absorption range used on UV-vis measurements was in the visible spectrum, and we used a universal polystyrene cuvette of 0.8 mm.

The DC electrical conductivity (σDC) was calculated from impedance measurements according to the methodology described by Heilmann (2003) [23]. Dielectric measurements in the frequency range from 40 Hz to 110 MHz were carried out with an Agilent Precision Impedance Analyzer 4249A (Santa Clara, CA, USA). The amplitude of the measuring signal was 100 mV. The nanocomposite was previously dried under a vacuum to eliminate the moisture, and measurements were conducted in a vacuum cell [24].

### 2.3. Cellular Viability

Six-week-old Balb/c mice were sacrificed by cervical dislocation, and resident peritoneal macrophages were obtained by repeated washings and retrieval of the peritoneal cavity with 10 mL of sterile and cold RPMI-1640. On the other hand, peripheral blood mononuclear cells (PBMC) were isolated from peripheral blood obtained from regular human donors; the blood was diluted with phosphate buffer solution (PBS) at a ratio of 1:1 (vol/vol) and then centrifuged on a Ficoll-Paque gradient for 30 min at 500× *g* at room temperature; the interphase layer consisting of PBMC was washed three times with a culture medium. The HEPG2 cancer cell line derived from human liver hepatocellular carcinoma was purchased from ATCC (American Tumor Cancer Collection, Manassas, VA, USA) and cultured in DMEM medium supplemented with 10% fetal bovine serum (FBS) and 1% antibiotic-antimycotic solution (GIBCO, Grand Island, NY, USA), in an atmosphere at 37 °C and 5% CO_2_. The cells were adjusted at 5 × 10^3^ cells/mL, seeded by triplicate on the different nanocomposites previously poured in 96-well plates (Corning, Corning, NY, USA) and incubated for 72 h in an atmosphere at 37 °C and 5% CO_2_. The cellular viability was evaluated by MTT or trypan blue staining according to the manufacturer’s instructions [25].

### 2.4. Nitric Oxide Determination

The supernatants of each treatment were used to determine the nitric oxide production by nitrate–nitrite colorimetric assay kit (Cayman Chemical, Ann Arbor, MI, USA) according to the manufacturer’s instructions [26]. Briefly, 40 μL of supernatants were mixed with 40 μL of assay buffer, 10 μL of enzyme cofactor, and 10 μL of nitrate reductase and incubated at room temperature for 3 h (to allow the conversion of nitrate to nitrite); the samples were analyzed by triplicate. After 10 min of incubation in Griess reagent at room temperature, the absorbance was measured at 560 nm in a microplate reader (Winooski, VT, USA, model EL311).

### 2.5. Evaluation of Cytokine Expression by Flow Cytometry

The cytokine production was analyzed by a BD cytometric bead array (CBA) Th1/Th2/Th17 CBA mice cytokines kit (BD, San Diego, CA, USA). For the evaluation, spleens were taken from mice 48 h after wounding. The spleens were crushed and homogenized in PBS (500 μL); the supernatants of this mixture were collected by centrifugation at 1600 rpm/10 min and stored at −20 °C until analysis according to the manufacturer’s instructions. The cytokines evaluated were different interleukins (IL) as IL-2, IL-4, IL-6, IL-10, IL-17A, INFΥ, and TNF. The cytokine production was measured by flow cytometer (BD Accuri C6, BD Biosciences, San Jose, CA, USA) according to the manufacturer’s instructions. CBA analysis was performed using FCAP array v1.0 software (Soft Flow Inc., St Louis Park, MN, USA).

### 2.6. Wound Healing Effect

Wound healing effects on mouse skin were investigated using nanocomposites CTS-g-GMA, (CTS-g-GMA)-AuNPs, and ((CTS-g-GMA)-AuNPs-Col) with different HAuCl_4_ concentrations (0.15, 0.3, 0.6, 1.07, 2 and 3 mM). A commercial material TheraForm^®^ (natural wound healing matrix of highly purified atelocollagen) (Surgical Esthetics^TM^, Northridge, CA, USA) was used as a positive control. First, the films were cut into squares with dimensions of 8 × 8 mm^2^; afterward, square films were washed twice in 600 μL of PBS for 5 min and then washed three times in 600 μL of DMEM for 5 min. Finally, the materials were sterilized by ultraviolet light (CL-100 UV) for 10 min.

All experimental protocols were approved by the ethics research and animal wellness committee (N◦CEIBA-2013-013) of the Facultad de Ciencias Biológicas, Universidad Autonóma de Nuevo León, and were done according to NOM-062-ZOO-1999 (Mexican legislation). Experimental animals were treated according to the criteria described in the PHS Policy on Humane Care and Use of Laboratory Animals and the “Guide for the Care and Use of Laboratory Animals” (NIH publication 86-23). Six-week-old female Balb/C mice (22 g body weight) were housed in polycarbonate cages at room temperature (21 ± 2 °C) on a 12-h light–dark cycle. Each was anesthetized intramuscularly with ketamine hydrochloride 100 mg/kg and xylazine 5 mg/kg. [27]. After that, the backs of the animals were shaved and swabbed with 70% ethanol three times before wounding.

Two skin squares of 5 × 5 mm (dermis and epidermis) on the back were selected in the shaved area. The wound nearby the head of the mice was implanted with the nanocomposites or controls; the other wound nearby the tail was used as a negative control without material, as shown (Figure 1). After surgery, lesions were evaluated daily for 40 days according to established parameters by the manual of wounds and ulcers and the OECD 404 guide; 3 specimens per material were used according to the requirements of the animal ethics committee [28] (Appendix A).

### 2.7. Cytokine Production

Mice had the nanocomposite surgically implanted; after 48 h, 2 mL of blood was extracted and they were sacrificed. The blood was centrifuged at 1000 rpm to obtain serum, which was frozen at −20 °C until analysis.

Additionally, each animal had its spleen extracted, which was homogenized with 200 mL of PBS sterile and centrifuged at 1600 rpm per 10 min to obtain the supernatant; it was frozen at −20 °C until analysis.

The cytokines were analyzed using a flow cytometer model Accuri C6 (Biosciences Center, North Brunswick Township, NJ, USA) following the protocol of BD cytometric bead away Mouse Th1/Th2/Th17 Cytokine Kit, which was used to detect IL-2, IL-4, IL-6, IFN-g, TNF, IL-17A, and IL-10 in all the samples.

### 2.8. Statistical Analysis

All experiments were performed in triplicate and statistical analysis was performed using analysis of variance (ANOVA). The results were considered statistically significant if the *p*-value was < 0.05.

## 3. Results

### 3.1. Size and Stability of AuNPs

The UV-vis spectra of the CTS-AuNPs in a liquid medium, synthesized with different concentrations of HAuCl_4_ (Figure 2), show absorption band maxima in the range of 516 and 522 nm. The slight redshift may be attributed to an increase in the average size of the particles due to the formation of clusters.

Figure 3 shows a TEM micrograph of the CTS-AuNPs synthesized in a liquid medium with the minimum (0.3 mM) and maximum (1.72 mM) precursor concentrations. The average size was from 8 to 20 nanometers. Figure 4 shows an SEM micrograph of films with gold nanoparticles, with a concentration of 0.3 (a), 0.6 (b), and 1.07 (c) mM, embedded into the CTS-based matrix.

At a low concentration of HAuCl_4_, the distribution of the nanoparticles was homogeneous in the matrix and no clustering was observed. A further increase in nanoparticle concentration (1.7 mM) resulted in the formation of some clusters.

### 3.2. Infrared Spectroscopy

Figure 5 shows the IR spectra of the different materials of CTS, CTS-AuNPs, CTS-g-GMA-AuNPs, and CTS-g-GMA-AuNPs-Col. The spectra permit observation of the difference between all materials. The relative intensity of the broadband centered at about 3300 cm^−1^ corresponds to the stretching of the O-H group, and bands at 1070 and 1020 cm^−1^ related with asymmetric stretching vibrations of the ether group of methacrylate decreased in materials with GMA. Therefore, we can assume that there is no effect or presence of acetic acid in the nanocomposite due to previous work [29].

The relative intensity of the primary amide group related to stretching vibration of C=O bonds at 1650 cm^−1^ decreased, especially for (CTS-g-GMA)-Col and ((CTS-g-GMA-AuNPs)-Col nanocomposites. The bending vibration of NH_2_ at 1550 cm^−1^ showed a decrease in relative intensity and a redshift for composite samples compared with pure CTS. Additionally, observed variations of bands at 1400 and 1421 cm^−1^ produced by stretching of C-N groups increased, along with vibrations of CH_2_-CO. The characteristic absorption bands of pure CTS are presented in Table 1. Several have authors reported FTIR spectra of collagen type I and GMA [30,31,32,33,34].

### 3.3. X-ray Diffraction (XRD)

Pure CTS is a semicrystalline biopolymer and has an XRD pattern with broad peaks at around 2θ° = 15° and 20°, indicating the average intermolecular distance of the crystalline part of pure CTS crystallizing in an orthorhombic unit cell [35]. Furthermore, the reflection at 14.8 and 15 degrees, with the diffraction line at 20 degrees, of the nanocomposite with gold nanoparticles and the pure CTS, respectively, is due to the absence of moisture in the material and the crystal form type I [36,37].

Figure 6a shows the X-ray diffraction of pure chitosan (CTS), CTS-g-GMA, and (CTS-g- GMA-AuNPs)-Col. The main differences in the XRD patterns are that the broad diffraction line at around 20 degrees ((200) reflection) in the 2θ scale increases in intensity and decreases in position in composites, compared to the pure CTS. The reflection centered at around 20 degrees has contributions from the amorphous phase, but the diffraction line at 15 degrees corresponds to the crystal form type I (Figure 6). In contradistinction to the diffractogram of the pure CTS and the nanocomposite of (CTS-g-GMA-AuNPs)-Col, the material of CTS-g-GMA without the gold nanoparticles is completely amorphous due to the only diffraction line at 20 degrees [37].

The diffraction line at about 38 degrees corresponds to the (111) planes in the cubic structure of the AuNPs, which can only be seen in the (CTS-g-GMA-Au)-Col nanocomposite [38,39]. In Figure 6b, the intensity of this line increases with the concentration of AuNPs.

### 3.4. Conductivity Measurements

According to practical media theory, the DC conductivity of a system with conductive inclusions in a dielectric matrix must increase; all samples were dried under a vacuum [27]. In our results, the incorporation of AuNPs results in a decrease in the conductivity for concentrations below 0.5 mM; a further increase in AuNPs concentrations increases the conductivity. For concentrations above 5.0 mM, the conductivity has an exponential increase of more than 12 orders of magnitude. The initial decrease in conductivity can be explained due to the interaction of AuNPs with the CTS matrix (Figure 2) and, therefore, the number of free H^+^ and OH^−^ ions, which are known to be responsible for the intrinsic ionic conductivity in pure CTS [24]. The exponential increase in conductivity with the increase in nanoparticle concentration is due to the percolation of the AuNPs nanoparticles with an increasing number of conductive paths in the composite matrix. This minimum appears due to the competition of two mechanisms: a decreasing number of free H^+^ and OH^−^ ions and the increasing metallic conductance of the embedded nanoparticles (Figure 7).

### 3.5. Cellular Viability

The nanocomposites containing different concentrations of gold nanoparticles (0.3, 0.6, 2, and 3 mM) did not show an effect on the cellular viability of murine peritoneal macrophages and PBMC (*p* < 0.05) compared with untreated cells (0%) (Figure 8). On the other hand, the different nanocomposites significantly affected (*p* < 0.05) the HEP-G2 cell viability at 0.3 (30%), 0.6 (20%), 2 (16%), and 3 (16%) mM, compared with untreated cells (0%).

### 3.6. Nitric Oxide

The different nanocomposites evaluated did not induce (*p* < 0.001) nitric oxide production in the murine peritoneal macrophages and HEP-G2 cells, compared with the lipopolysaccharide (LPS) treatment, which induced inflammatory response and high nitric oxide production (Figure 9).

### 3.7. Cytokine Production by Flow Cytometry

In this study, some materials affected the inflammatory response in physiological conditions by affecting pro-inflammatory (TNF, IL-2, and IL-17A) and anti-inflammatory cytokines (IL-4 and IL-10). In addition, IL-6 is a pro and anti-inflammatory cytokine [40,41,42].

In this analysis, TheraForm^®^ was used as a control to induce cytokine production. Compared with the commercial product, the CTS-g-GMA material induced significantly (*p* < 0.05) more production of IL-2 and IL-4 cytokines, but no significant difference was found for other nanocomposites. IL-6 production was induced significantly (*p* < 0.05) more for CTS-g-GMA, (CTS-g-GMA)-AuNPs 0.15 mM, and ((CTS-g-GMA)-AuNPs)-Col 0.15 mM, and highly significantly (*p* < 0.001) more for ((CTS-g-GMA)-AuNPs)-Col 0.6 mM, (CTS-g-GMA)-AuNPs 2 mM, and (CTS-g-GMA)-AuNPs 3 mM. A significant difference was not found for the other materials. INFΥ production was induced significantly (*p* < 0.05) more in the mice treated with CTS-g-GMA and (CTS-g-GMA)-AuNPs 0.15 mM (Table 2).

Additionally, a difference was found in the production of TNF for CTS-g-GMA, (CTS-g-GMA)-AuNPs 0.15 mM, ((CTS-g-GMA)-AuNPs)-Col 0.15 mM, ((CTS-g-GMA)-AuNPs)-Col 0.6 mM, (CTS-g-GMA)-AuNPs 1.07 mM, and ((CTS-g-GMA)-AuNPs)-Col 1.07 mM y (CTS-g-GMA)-AuNPs 2 mM, whereas for others materials, a significant difference compared with TheraForm^®^ was not found (Table 2).

The production of IL-17A was significant (*p* < 0.05) for CTS-g-GMA, (CTS-g-GMA)-AuNPs 0.15 mM, and ((CTS-g-GMA)-AuNPs)-Col 0.15 mM. Additionally, a significant difference was found between mice treated with CTS-g-GMA, CTS-g-GMA-AuNPs 0.15 mM, and ((CTS-g-GMA)-AuNPs)-Col 0.15 mM compared with TheraForm^®^ (Table 2). Furthermore, the expression of cytokines in the spleen did not show a significant difference (*p* < 0.05) between the evaluated nanocomposites and TheraForm^®^ (Table 3).

### 3.8. Wound Healing Test

All the nanocomposites and TheraForm^®^ induced a process of recovery at seven days in the mice, with healthy classified as score wound type I (Appendix A), characterized by erythematous aspect, lesion extension < 1 cm, lesion deep < 1 cm, exudate quantity and quality absent, necrotic tissue absent, granular tissue 75–100%, edema absent, pain 0–1, and surrounding skin healthy.

On day one, the recovery percentage in in terms of wound closure was not different between treatments. However, on the third day, the treatments with ((CTS-g-GMA)-AuNPs)-Col 0.3 and 0.6 mM were accelerated (25% both) compared to TheraForm^®^ and another nanocomposite (12.5%).

On the seventh day, the ((CTS-g-GMA)-AuNPs)-Col 1.07 mM nanocomposite induced 62.5% wound closure compared with nanocomposites that contained a lesser amount of gold nanoparticles concentrations—0.6, 0.15, and 0.3 mM (50%, 37.5%, and 37.5%, respectively). It even induced a better percentage than the commercial product, TheraForm^®^ (37.5%). The absence of gold nanoparticles in the nanocomposite, CTS-g-GMA, decreased the process of wound closure (25%). Similar data were observed at 18 days. (Figure 10).

On day three, the nanocomposites ((CTS-g-GMA)-AuNPs-col) at AuNP concentrations of 0.3 and 0.6 exhibited better wound healing percentages (*p* < 0.05). On day eight, the nanocomposite with the highest concentration of AuNPs (1.07) induced superior wound healing percentages, followed by those nanocomposites with AuNP concentrations of 0.6 and 0.3, respectively (*p* < 0.05). Furthermore, we found no significant difference (*p* < 0.05) between composites with 0.3 AuNP containing or not collagen (day 8).

On day 18, the addition of collagen to the composites showed no significant differences (*p* < 0.05) concerning wound healing between groups containing the same AuNP concentration. However, at the highest AuNP concentration (1.07), the wound healing was greatest (*p* < 0.05). When compared with Theraform (positive control of wound repair), the only significant difference (*p* < 0.05) was found with a composite of 0.15. On day 40, no significant difference was found between any group (*p* < 0.05).

Other works using different polymer matrices and gold nanoparticles have had similar results; the nanoparticle’s shape, surface modification, and synthesis method have a crucial effect on wound healing [7,43]. After 21 days, all treatments showed total recovery and cicatrization. The ((CTS-g-GMA)-AuNPs)-Col treatment showed better cosmetic appearance, such as less deep skin pigment. In general, all wounds, treated and untreated, allowed the growth of new skin without infection and inflammatory signs during the wound healing process for the 40 days evaluated (Figure 11).

### 3.9. Histological Tests

We tested our nanocomposites for hematoxylin and eosin staining in the wound area of the murine tissue. In Figure 12, we show the response to both stainings for CTS-g-GMA (a, c) and ((CTS-g-GMA)-AuNPs) 0.3 mM (b, d) at 15 and 30 days after surgery. Here, we observe the histological sections: red arrows denote the epidermis section; black arrows denote the neoformation of pilose follicles; blue arrows denote the neoformation of blood vessels.

We found the best results for ((CTS-g-GMA)-AuNPs) 0.3 mM, with the greatest epidermis and blood vessel reconstitution.

## 4. Discussion

For application in tissue engineering, a material should meet the following requirements: biocompatibility with the tissues, biodegradability, nontoxicity, and suitable mechanical properties. The CTS-AuNP nanocomposite responds to all these requirements but has relatively insufficient mechanical strength. However, Elizalde et al. showed that with a correct molar relation, this hydrogel can change the mechanical properties, viscosity, and swelling index of CTS and can be used for cellular support for biomedical applications, allowing the adhesion and growing of human fibroblasts [12].

The interaction between CTS and gold nanoparticles was demonstrated by infrared spectroscopy; the relative decrease in the intensity of the band at 1550 cm^−1^ with the increase in the gold concentration has been attributed to interactions between NH_3_^+^ and the polarizer surface of the nanoparticle for nanocomposites of silver and gold using CTS as reducing agent [18,19].

The conductivity of the nanocomposites strongly depends on the HAuCl_4_ concentration and moisture. Materials with concentrations above 0.5 mM showed the formation of nanoparticle clusters (Figure 5), producing percolation of nanoparticles followed by an increase in the number of conductive paths; additionally, the interaction of -OH groups and protonated amine can be corroborated by thermogravimetric analysis reported previously by our group [24]. All nanocomposites with collagen showed better results than controls and (CTS-g-GMA)-AuNPs; this is because the collagen type I used aids the process of scar tissue, which is the main component of the intact uninjured dermis [44].

Despite this, (CTS-g-GMA)-AuNPs at 2 and 3 mM increased the production of cytokines IL-6. They decreased levels of IL-17A; it has been reported that this concentration of cytokines promotes the healing of wounds [45]. These materials cannot be considered the best materials for re-epithelialization because they are breakable and fragile.

Any material used for skin wound healing must not permit pathogen infection or allergic response, because bacterial components, infections, or allergic materials may impair the repair mechanism of the host by interfering with cell–matrix interactions or attenuating the inflammatory response [46]. The gold nanoparticles show an excellent antibacterial capacity that changes in response to the shape and surface modifications of the nanoparticle [43,47]. All evaluated nanocomposites induced the cicatrization process without signals of fever, infection, allergy, or toxicity.

Wound healing is a dynamic process involving three phases overlapping in time and space: inflammation, tissue formation, and tissue remodeling; inflammatory response is the first of the several overlapping processes that constitute wound healing. The addition of collagen led to a higher percentage of healing at 3 and 8 days, compared with the commercial control and materials without collagen. The nanocomposites ((CTS-g-GMA)-AuNPs)-Col at 0.3 and 0.6 mM showed better performance at early healing times. The inflammatory response supplies growth factor and cytokine signals that orchestrate the cell and tissue movements necessary for repair [46]. The positive control (TheraForm^®^) presented low production of cytokines (IL-2, IL-4, IL-6, INFΥ, TNF, IL-10) and an absence of IL-17A. We expected to obtain similar results from the nanocomposites evaluated; nevertheless, differences were observed in the production of cytokines in a material compound-dependent manner.

It is impossible to know the average production of cytokines per individual, and it would be a mistake to generalize; this is considered in our model, which compared wound healing between each individual and each group, including two injuries per mice. After injuries, the IL-1b, INF-g, and TNF at the wound site led to the expression of various classes of adhesion molecules essential for leukocytes adhesion and diapedesis (P-and E-selectins as well ICAM 1, -2, LFA-1, MAC-1, gp150, chemokines, and their receptors, including IL-8, growth-related oncogene-alpha, and monocyte chemoattractant protein-1).

The early stages (1 to 3 days) are the most critical factor in the tissue regeneration process due to the transcription factors and proteins produced by cells to activate the process of cell migration [48]. The ((CTS-g-GMA)-AuNPs)-Col nanocomposite at 0.6 mM was considered the best material to induce healing wounds in less time, along with an aesthetic, functional aspect, by modulating the cytokine production between the inflammatory and anti-inflammatory cytokines. Successful skin repair after tissue injury requires resolution of the inflammatory response. (CTS-g-GMA)-AuNPs at 1.07 mM with and without collagen showed an increase in wound healing percentage at 8 days; this stage is related to the second phase of wound healing, or granulation process [49].

Whereas knowledge about mechanisms and molecules inducing and perpetuating the inflammatory response is constantly increasing, mechanisms that limit and downregulate this activity are less appreciated (down-regulation of chemokine expression by anti-inflammatory cytokines such as IL-10 or TGF-B1, or up-regulation of anti-inflammatory molecules such as IL-1 receptor antagonists or soluble TNF receptors). For example, McFarland et al. showed that mice with impaired capacity macrophages have an excess of fibrin and cellular debris, delaying wound healing compared with untreated mice; in the same way, it was observed that in IL-6 knockout mice, the wound healing was delayed because of poor infiltration of macrophages, affecting the re-epithelialization [50]. Additionally, IL 17 and IL-10 knockout mice showed a faster wound healing process because the inflammatory process was decreased, which improved the angiogenesis [51,52].

The production of spleen cytokines was not different compared with TheraForm^®^, suggesting that these materials cannot produce a systemic inflammatory response that can compromise the individual’s homeostasis. For future studies with the best nanocomposite to induce the best aesthetic cicatrization, we are considering examining matrix metalloproteases, because these molecules play a role in the process of tissue granulation, modulating the formation of scar tissue by the production of extracellular matrices and promoting cell migration, which is reduced and changed by type III collagen and replaced by type I collagen [45].

## 5. Conclusions

The chitosan-based Au nanocomposites presented here (with and without type I collagen) exhibit a superior ability to promote skin wound healing when compared to the positive controls TheraForm^®^ and Chitosan-g-glycidyl methacrylate/Au nanocomposites ((CTS-g-GMA)-AuNPs) (with and without type I collagen). The best healing process was achieved using nanocomposites with 0.6 mM of gold nanoparticles. Our results demonstrated that ((CTS-g-GMA)-AuNPs)-Col is an efficient material for accelerating wound healing by improving the formation of granulation tissue and the migration of hyperproliferative keratinocytes; these observations are traceable to the nanocomposite’s electrical behavior according to the electrical percolation phenomena. In this regard, our results suggest that the best performance nanocomposite is found with a nanoparticle concentration near the percolation threshold. The percolation threshold dictates when nanoparticles agglomerate, and the nanocomposite starts to fail. It is noteworthy that further studies are required to correlate cytokine production and the cicatrization process for a better understanding of this mechanism.

## Figures and Tables

**Figure 1 pharmaceutics-14-01855-f001:**
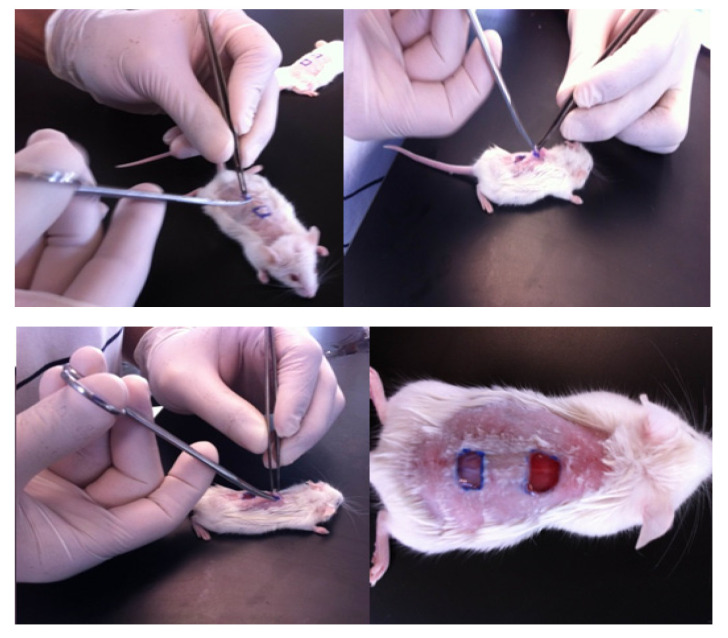
Surgery process. The lesion nearby the head was implanted with nanocomposites or TheraForm^®^ as a positive control, and the location near the tail was an untreated wound as a basal control for wound healing.

**Figure 2 pharmaceutics-14-01855-f002:**
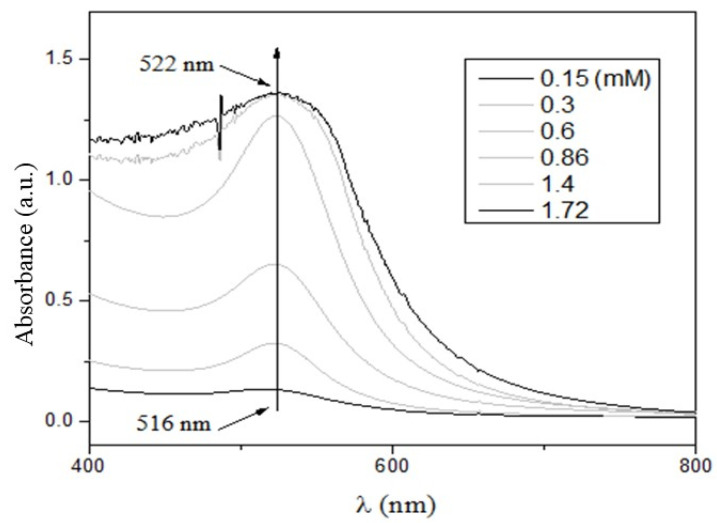
UV-vis spectrum of the liquid samples of gold nanoparticles synthesized with different concentrations.

**Figure 3 pharmaceutics-14-01855-f003:**
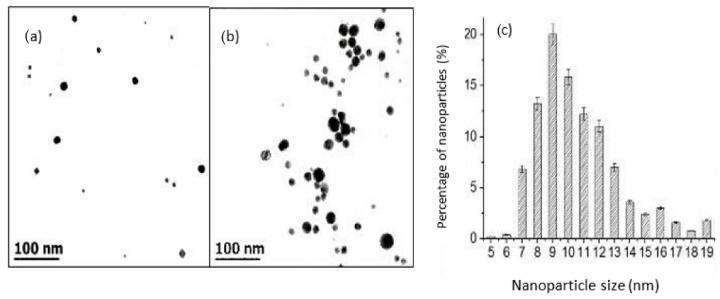
Transmission electron microscopy of CTS-AuNPs obtained from solution with (**a**) 0.3 and (**b**) 1.07 mM of HAuCl_4_; (**c**) the particle size distribution histogram of CTS-Au-NPs obtained from TEM micrographics.

**Figure 4 pharmaceutics-14-01855-f004:**
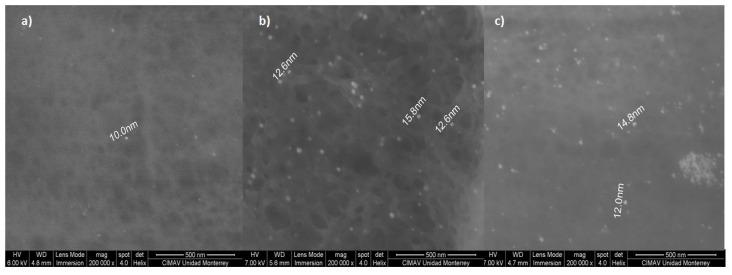
Scanning electron microscopy micrographs of CTS-g- GMA-Au-Col nanocomposites: (**a**) 0.3 mM, (**b**) 0.6 mM, (**c**) 1.07 mM. The increment of nanoparticles in the material induces the formation of clusters.

**Figure 5 pharmaceutics-14-01855-f005:**
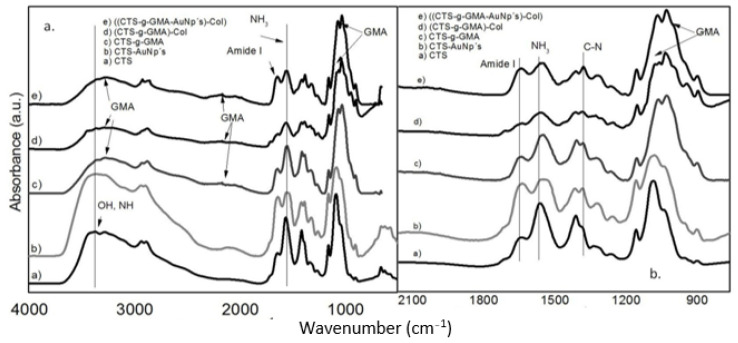
(**a**) FTIR spectrum obtained for pure chitosan (CTS), nanocomposites with chitosan and gold nanoparticles CTS-AuNPs; chitosan, and glycidyl methacrylate (CTS-g-GMA); chitosan, glycidyl methacrylate, and type I collagen (CTS-g-GMA)-Col; and a nanocomposite (CTS-g-GMA-AuNPs)-Col; (**b**) zoom of the region of greatest contribution between 1700 to 900 cm^−1^.

**Figure 6 pharmaceutics-14-01855-f006:**
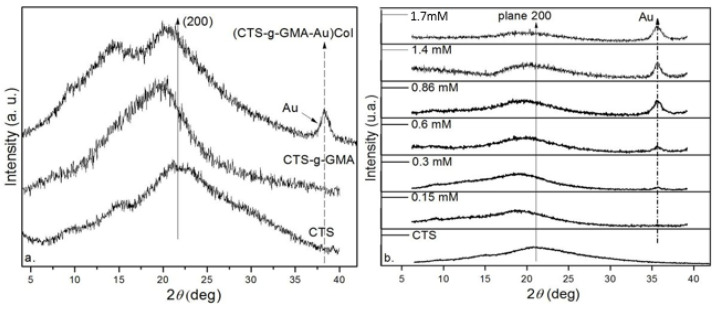
(**a**) CTS diffractograms compared with CTS-g-GMA and (CTS-g-GMA-AuNPs)-Col. (**b**) X-ray diffraction of nanocomposites of chitosan and gold nanoparticles with different concentrations of HAuCl_4_.

**Figure 7 pharmaceutics-14-01855-f007:**
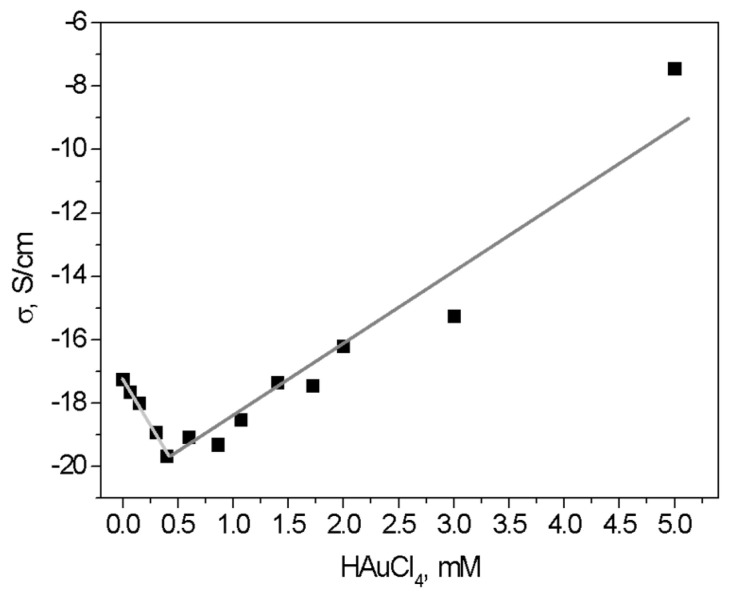
Dependence of DC conductivity of films at room temperature on the concentration of HAuCl_4_ (lines a guide to the eye).

**Figure 8 pharmaceutics-14-01855-f008:**
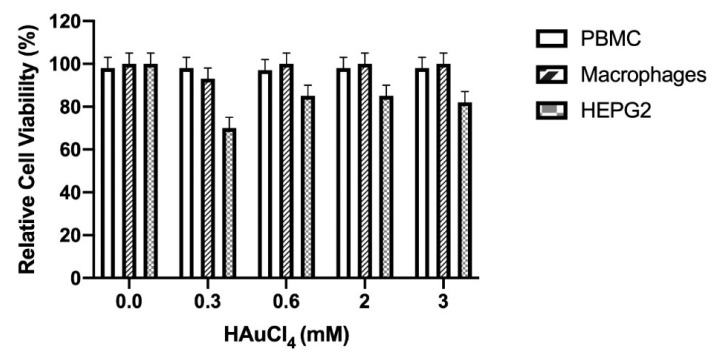
Relative cell viability of PBMC, macrophages, and HEP-G2 cells in the presence of concentrations of CTS-AuNPs in a liquid medium (0.3, 0.6, 2, and 3 mM); *n* = 3, SD ± 0.5, *p* < 0.05.

**Figure 9 pharmaceutics-14-01855-f009:**
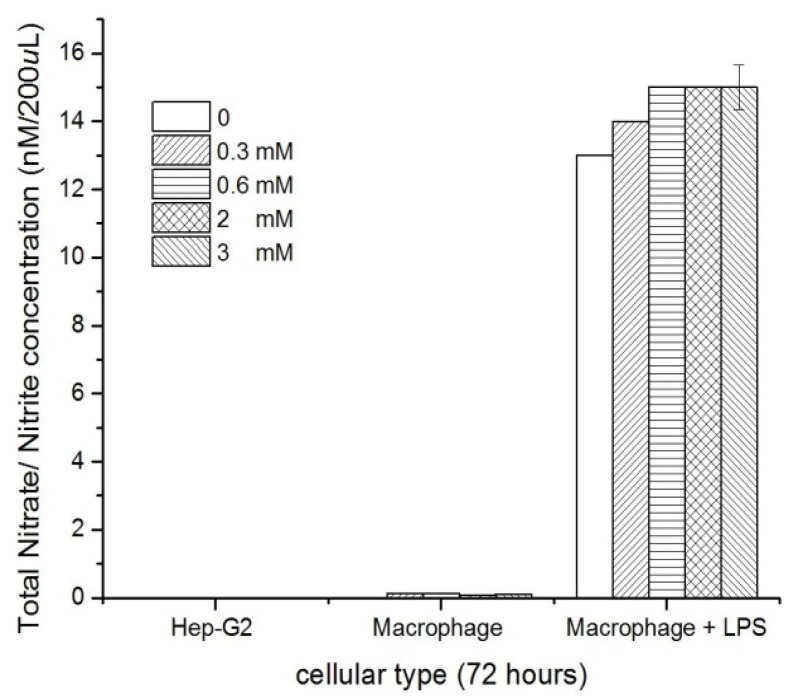
Nitric oxide production of HEP-G2 and macrophages due to exposure to CTS-AuNPs in a liquid medium at different concentrations, in comparison with macrophages under an LPS treatment with different concentrations of CTS-AuNPs, *n* = 3, SD ± 0.5, *p* < 0.001.

**Figure 10 pharmaceutics-14-01855-f010:**
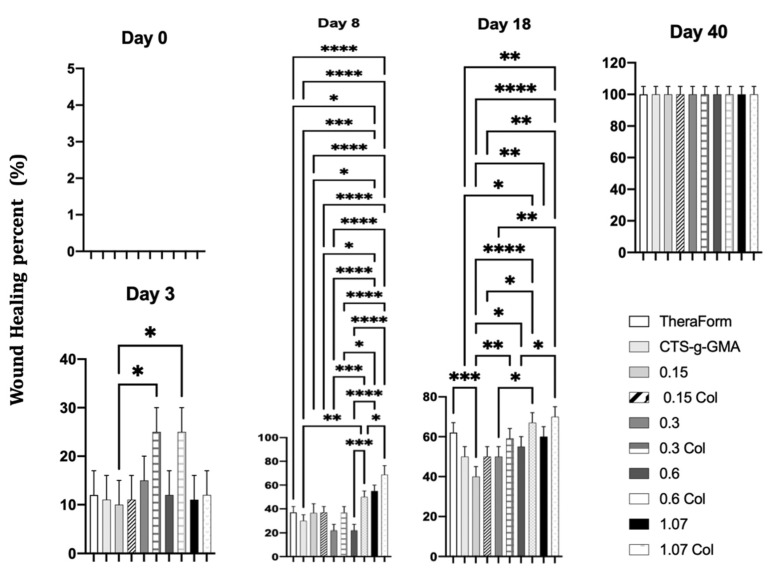
Wound healing percentage after 40 days, with TheraForm^®^; CTS-g-GMA; (CTS-g-GMA)-AuNPs at different concentrations—0.15, 0.3, 0.6, 1.07 mM; and ((CTS-g-GMA)-AuNPs)-Col at different concentrations—0.15, 0.3, 0.6, 1.07 mM. (*p* < 0.05), *n* = 3, SD ± 0.5. The asterisk means that there are note significant differences between that samples.

**Figure 11 pharmaceutics-14-01855-f011:**
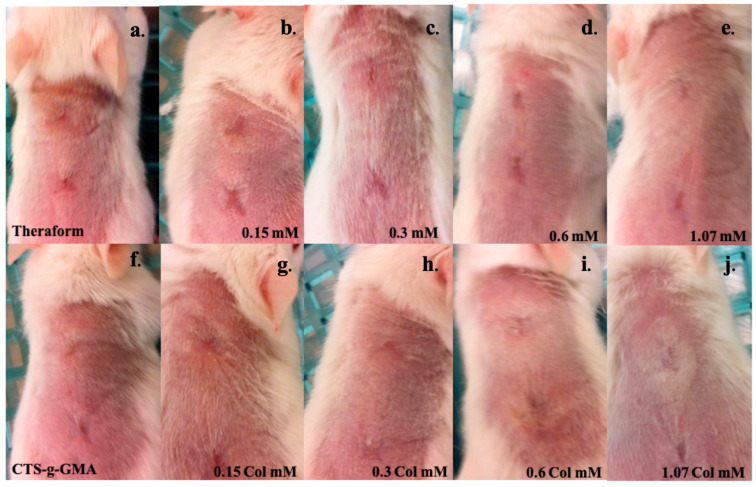
Skin recovery after 40 days of treatment with (**a**) TheraForm^®^; (**f**) CTS-g-GMA; (CTS-g-GMA)-AuNPs at different concentrations: (**b**) 0.15 mM, (**c**) 0.3 mM, (**d**) 0.6 mM, (**e**) 1.07 mM; and ((CTS-g-GMA)-AuNPs)-Col at different concentrations: (**g**) 0.15 mM, (**h**) 0.3 mM, (**i**) 0.6 mM, (**j**) 1.07 mM.

**Figure 12 pharmaceutics-14-01855-f012:**
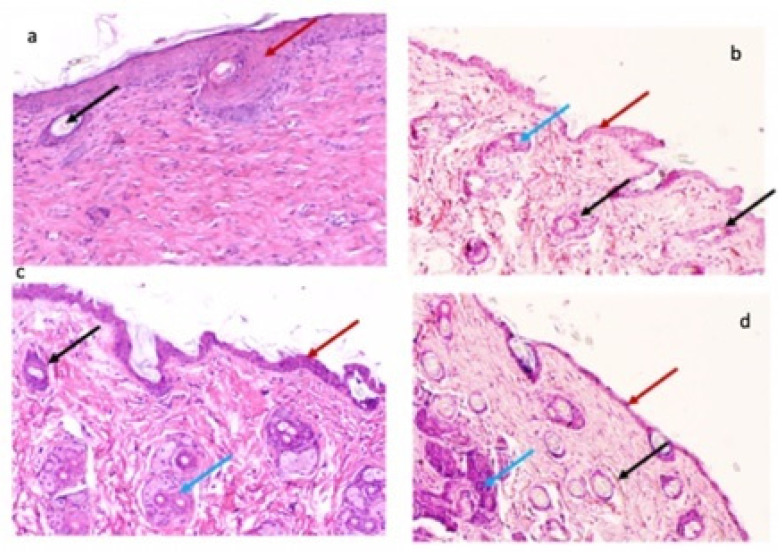
Hematoxylin and eosin staining of tissue wound area. Murine tissue treated with CTS-g-GMA (**a**,**c**) and ((CTS-g-GMA)-AuNPs) 0.3 mM (**b**,**d**) at 15 and 30 days after surgery. The red arrows show the epidermis, black arrows show hair follicles, and blue arrows indicate the formation of new blood vessels. Magnification: 10× to improve visualization.

**Table 1 pharmaceutics-14-01855-t001:** Characteristic functional groups of pure CTS.

Functional Group	Peak Position (cm^−1^)
OH (stretching S.)	3450
NH (stretching S.)	3360
CH_2_ (stretching S.y as glucopyranose)	2870, 2912, 1430
C=O (stretching As amide I)	1650
NH_2_ (deformation As.)	1560–1590
C=O	1730
CH_2_ (amide II)	1420
CH_3_ and C-CH_3_ (amide III)	1380
C-O	1255
C-O-C (glycosidic bond)	1040–1150
C-H	838–850
OH (stretching S.)	3450

**Table 2 pharmaceutics-14-01855-t002:** Induction of serum cytokines due to exposition with Au nanocomposites and collagen.

Serum Cytokines (pg/mL)							
Treatment [mM]	IL-2	IL-4	IL-6	INFϒ	TNF	IL-17A	IL-10
TheraForm^®^	0.28	2.87	2.04	2.97	2.12	0	1.84
CTS-g-GMA	3.94 *	9.5 *	26.15 *	11.1 *	39.84 *	4.93 *	31.12 *
((CTS-g-GMA)-AuNPs [0.15]	0.53	2.47	17.27 *	17.3 *	6.4 *	1.74 *	7.5 *
((CTS-g-GMA)-AuNPs)-Col [0.15]	0.32	3.3	11.2 *	4.64	13.59 *	0.93 *	14.67 *
((CTS-g-GMA)-AuNPs [0.3]	0.47	2.23	20.54 *	4.73	5.32	0.4	16.3 *
((CTS-g-GMA)-AuNPs)-Col [0.3]	0.32	0	15.8 *	4.98	15.6 *	1.55 *	13.7 *
((CTS-g-GMA)-AuNPs [0.6]	0.45	0	2.26	5.46	4.38	0.21	1.21
((CTS-g-GMA)-AuNPs)-Col [0.6]	0	0	94.06 **	2.34	9.86 *	0.65	0
((CTS-g-GMA)-AuNPs [1.07]	0	0	2.44	0.83	5.59 *	0	0
((CTS-g-GMA)-AuNPs)-Col [1.07]	0	0	2.63	0.87	7.06 *	0	0
((CTS-g-GMA)-AuNPs [2]	0	0	944.26 **	2.44	27.21 *	0.34	0
((CTS-g-GMA)-AuNPs [3]	0	0	174.58 **	2.98	1.78	0.11	0

*p* < 0.05 *; *p* < 0.001 **.

**Table 3 pharmaceutics-14-01855-t003:** Induction of spleen cytokines due to exposition with Au nanocomposites and collagen.

Spleen Cytokines (pg/mL)							
Treatment [mM]	IL-2	IL-4	IL-6	INFϒ	TNF	IL-17A	IL-10
TheraForm^®^	1.79	0	0.63	7.3	19.7	0	0
CTS-g-GMA	0.67	0	0.48	3.63	10.08	0	1.32
((CTS-g-GMA)-AuNPs [0.15]	1.23	0	1	8.58	19.95	0	3.23
((CTS-g-GMA)-AuNPs)-Col [0.15]	0.8	0	0.78	3.31	9.67	0.04	0
((CTS-g-GMA)-AuNPs [0.3]	1.46	0	0.92	4.86	22.2	0	0
((CTS-g-GMA)-AuNPs)-Col [0.3]	0.75	0	0	3.57	9.83	0.16	0
((CTS-g-GMA)-AuNPs [0.6]	0.9	0	0.88	3.42	29.52	0	0
((CTS-g-GMA)-AuNPs)-Col [0.6]	0.68	0	0	2.76	9.41	0.11	0
((CTS-g-GMA)-AuNPs [1.07]	0.71	0	0	4.9	7.91	0	4.91
((CTS-g-GMA)-AuNPs)-Col [1.07]	0.7	0	0.48	4.71	14.11	0	0
((CTS-g-GMA)-AuNPs [2]	1.25	0	1.35	2.06	7.56	0.16	0
((CTS-g-GMA)-AuNPs [3]	0.32	0	0	6.16	8.35	0	0

## Data Availability

Not applicable.

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
