# Peer review of "Chitosan-G-Glycidyl Methacrylate/Au Nanocomposites Promote Accelerated Skin Wound Healing"

_pharmaceutics, 2022, doi:10.3390/pharmaceutics14091855_

Round 1

Reviewer 1 Report

The topic of the manuscript in interesting and the approach is very complex.

Some suggestions are listed below before publication:

Abstract

To replace “the cosmetic appearancein 3 positions in the abstract.

Introduction

To explain NADH to NAD

3.2 Size and stability of AuNPs

The average size was from 8 to 20 nanometers, which agrees with the UV-Vis spectroscopy/In my opinion, UV-Vis can just confirm the formation of Au nanoparticles but cant say anything about the  average size.

From Fig. 4c it is not clear what particles distribution is presented. The range of particle sizes is from 7 nm to 19 nm.

Figure 4. Transmission Electron Microscopy of CTS-AuNPs obtained from solution with (a) 0.3 and 270
(b) 1.07 mM of HAuCl4; (c) The particle size distribution histogram obtained from TEM micrographics.

Results

“After 21 days all treatment showed total recovery of cicatrization there is no difference in the total recovery of cicatrization be tween untreated and treated wounds”/in this case why you treat the wound if it is no difference between the treated and untreated wound? You have to mention the differences in the quality of cicatrization and the speed of healing here. I suggest the reformulation.

Conclusions

“Chitosan-based Au nanocomposites presented here (with and without type I colla gen) exhibit a superior ability for promoting skin wound healing when compared to the positive controls TheraForm® and CTS-g-GMA”/ I suggest to use Chitosan-g-glycidyl methacrylate/Au nanocomposites  ((CTS-g-GMA)-AuNPs) presented here (with and without type I colla gen)……

I suggest to insert some discussions on electrical behavior of nanocomposites before conclusions.

Reviewer 2 Report

Specific comments:

1.      Line 62: Strictly speaking, chitosan is not natural, but semi-synthetic polymer. Replace “natural polymer” to “polymer of natural origin” or “natural-origin polymer”.

2.      In my opinion, when describing the properties of nanocomposite films, it is quite odd and very inconvenient to operate with the amount of HAuCl4 expressed in mM (mmol/L) solution, 3 mL of which was added to the reaction mixture during the preparation of molding solutions to produce the composite films (lines 101-106). To make some sense, I would recommend replacing mmol/L HAuCl4 with mmol HAuCl4/g composite sample throughout the text.

3.      Line 102: The molecular weight (in contrast to the molar mass) is unitless. Either remove g/mol or use term molar mass across the text.

4.      Line 107: If I understand correctly, it should be a mass ratio, not a molar ratio.

5.      Line 133-135: Specify the thickness and material of the cuvette.

6.      The authors state that they used the ultraviolet-visible spectra to determine the sizes of the Au nanoparticles (lines 133-134) and these results correlate with the TEM data (line 255). However, I found neither the results themselves nor the formula for calculating the particle size in the manuscript.

7.      Lines 136-140: Specify the technique for measuring DC electrical conductivity of the nanocomposite films. Samples (of what dimensions? of what moisture content?) were placed in a cell (of which one?) with electrodes (of which one?).

8.      Section 2.3 Cellular viability: Add nanocomposite dimensions/mass to the methodology. Was there any pretreatment of the samples prior to culturing in the MTT test? The samples may have contained a variety of impurities (e.g. acetic acid, unreacted HAuCl4, etc.) that could have affected the results.

9.      It seems to me more logical to switch Sections 3.1 and 3.2. First, provide the characterization of nanoparticles in chitosan solution, and then the characterization of composite films.

10.  Section 3.1 Infrared spectroscopy: The authors need to rethink the description of the IR spectra, taking into account the fact that the samples may contain acetic acid (it is not completely removed by heating because it is bound to the amino group of chitosan by ionic bonding). The carboxyl group of acetic acid can appear in the IR spectrum both as a free acid (~1720 cm-1) and as a carboxylate anion at 1650-1550 cm-1 (asymmetric stretch) and 1400-1300 cm-1 (symmetric stretch).

In the authors' opinion, what is the decrease of the amide I band in (CTS-g-GMA)-Col and ((CTS-g-GMA-AuNPs)-Col composites (lines 233-235) associated with? Addition of collagen with more amide groups compared to chitosan (degree of acetylation 15%) should only increase the intensity of this band. It might make sense to add the IR spectrum of collagen for comparison.

11.  Section 3.2 Size and stability of AuNPs: By what does the reduction of HAuCl4 into Au particles occur? What is the reducing agent in this chemical process? It would be desirable to discuss this point here and, if possible, give an equation of the chemical reaction.

12.  Section 3.3 X-ray diffraction (XRD): The XRD section is quite weak. Add a more detailed description of X-ray diffractograms considering different chitosan polymorphs (See recent review on this topic DOI 10.1134/S1063774518030033).

13.  Line 306: What was the moisture content of the samples?

14.  Figures 8-10: Indicate the number of replications n, and make clear what error bars represent (standard error, standard deviation, confidence interval, etc.). In Figures 8 and 9, also indicate statistically significant differences.

15.  The English usage is often poor. Please check the paper once again to clarify meaning, improve awkward sentence structure and correct grammar.

16.  Check once again the appropriateness of using acronyms. Please note the following regarding acronyms:

Acronyms should be spelled out upon first use, followed by the acronym itself in parentheses.

Subsequently, only the acronym should be used in the text. E.g. the authors used ‘chitosan’ 33 times in the text, even though they introduced the acronym CTS.

To keep the use of acronyms to a minimum, only insert an acronym if the term is used at least three times.

17.  Across the text: use h for hours, and min for minutes.

Reviewer 3 Report

1.     How can functionalization can be done with synthetic and natural polymers for biomaterials to improve their properties is need to be explained.

2.     In figure 1, it is recommended to give the sub names and explain them with their titles in the fig 1 title.

3.     What is the situation if the wound happened while suffering from fever, is the author's approach can work effectively?

4.     The conditions and precautions or environments must be highlighted for wound healing.

5.     Conclusions should be improved by adding some important key results.   

Round 2

Reviewer 2 Report

The authors have successfully addressed all my concerns, improving the manuscript with their edits. In my opinion, the manuscript is now acceptable for publication. 

This manuscript is a resubmission of an earlier submission. The following is a list of the peer review reports and author responses from that submission.

Round 1

Reviewer 1 Report

In this work, the authors reported chitosan-based Au nanocomposites with superior healing performance on mouse skin wound than commercial product. This topic is of interest to the community. However, several minor issues should be addressed before acceptance of publication. They are listed below.  

1. Background in Figure 4b looks very dirty. Does this indicate the existence of very small particles? Also, the authors should use dynamic light scattering (DLS) instead of TEM to give the accurate particle size distribution. And the distribution percentage at large particle range (e.g. 1000 nm) should be shown in the chart to reveal there was no large precipitate.

2. Brightness in Figure 5 should be increased to make the difference clearer to readers.

3. The maximum range of Y axis in figure 8 should be adjusted to include the top of bars in the chart. Also, concentration of AuNPs in figure caption does not match with the values in the bar chart.

4. No footer was found in table 2 and 3. And the stds (standard deviations) of these cytokine measurements should be provided as well.  

5. Scale bar should be provided in Figure 12.

6. typo in line 467: skin would healing.

Author Response

First, we want to thank you for your comments regarding our work.

Line 179 was corrected, and you can find the correction highlighted in yellow.

References have been corrected and DOI numbers have been added.

Reviewer 2 Report

The submitted manuscript describes in a clear and comprehensive form the results of wound healing in applying a composite material with Au nanoparticles, chitosan, glycidic methacrylate, and collagen. The introductory section, the description the methods used, and the results are clearly described and discussed in detail. The results are compared with a commercially available product. This comparison is certainly welcome. The authors can be advised to concentrate on the procedure for the preparation of composite films in their future work, given the current time-consuming preparation time. There are no comments on the text and I agree to publish it as is.

Only on line 179 are characters of unknown meaning shown; they are probably correctly intended to be degrees Celsius. The references lack the indication of DOI numbers.

Author Response

We have unintentionally switched the responses of Reviewer 1. We are now attaching the responses to all Reviewers to correct the referred mistake.

Reviewer 3 Report

The manuscript Chitosan-g-glycidyl methacrylate/Au nanocomposites promotes accelerated skin wound healing by Héctor A López-Muñoz et al. describes the preparation of a series of hybrid composites that could be potentially utilized as wound-dressing materials. The materials were prepared and their physicochemical properties evaluated by means of XRPD, TEM, SEM, FTIR and electrical conductivity measurements. The biological effects exerted by the materials included first in vitro assessment of their cytotoxicity and induction/suppression of inflammatory factors, and next by applying materials in vivo to the wounds on the backs of the balb/c mice.

Such topic is important as wound healing process using both conventional and modern means sometimes fail and possible improvements are sought for. The manuscript roughly fits the scope of the journal Pharmaceutics. It is written rather well.

However, there are many drawbacks of the study that would have to be addressed before the article can be considered for publishing:

1.    The nanocomposites preparation section should be extensively rewritten. Currently, it is impossible to recreate the experimental conditions as there is data missing – for instance the preparation of CTS-AuNPs in lines 96-101 does indeed contain the concentrations of the solutions but not their volumes or ratios in which they were mixed making it pointless.

2.    Please rewrite the fragment in 102-106, as it is unclear preparation of what is described – CTS-g-GMA? Or is CTS-g-GMA further modified? Also – what was the purpose of potassium acetate in this procedure?

3.    “To achieve stable conditions…” (109). What were the stable conditions? How were they monitored and established?

4.    How much collagen was used in relation to the (CTS-g-GMA)-AuNPs to prepare the final materials? Stating only the concentration of the collagen is insufficient. Also – how much of the collagen was found to connect to the material? Was this tested anyhow? TGA-DSC would help.

5.    The gold nanoparticle content is never assessed nor confirmed. The amount of gold nanoparticles should be tested and reported.

6.    You mentioned that the impedance of the materials was measured, but no equipment is listed in lines 130-131. Also, the citation [18] is a 218 pages book – please provide at least the chapter from which the method was derived.

7.    The “cosmetic appearance” of the healed wounds should not be mentioned so many times (lines 33, 38, 364, 442). This is a subjective matter which cannot be objectively measured thus it does not belong in a scientific article.

8.    Wherever “manufacturer’s instructions” are given, please provide either a description of the method or a proper reference to either these instructions or another paper where such experimental details are given (see for example lines 137 or 164).

9.    “peripheral blood obtained from normal donors”(136-137). We are still talking about the mice? This is not stated in the paper. Also, please define normal donors. In this manner please also improve the 2.5 subchapter, as there is no information that the spleens were harvested post mortem. Additionally – was there one mouse per one material? It is not specified in the experimental.

10.  Concentrations of over 5.0 mM are mentioned (line 266) but such materials were never evaluated in any of the studies reported.

11.  Add error bars in figure 8.

12.  Tables 2 and 3 – why were the 0.3 mM composites not tested?

13.  Were the results presented in figure 10 tested for their significance? You do not discuss these results in terms of effect of collagen. You state that “nanocomposites with collagen showed better results than controls”(409), yet 0.15-col showed worse results, as well as only 0.3-col and 1.07-col showed better than the controls.

14.  You mention the mechanical properties of the materials, yet no evaluation of those was performed (i.e. tensile strength or Young’s modulus). To be able to discuss these properties additional test are required.

15.  You mention the antibacterial effect of the materials but such properties were not tested. The untreated controls did not show any bacterial infection as well, so this conclusion is exaggerated and based on other studies. The antibacterial effect could come from the CTS. You also mention the effect of visible light, but the gold nanoparticles would have to be internalized to the bacteria and then irradiated to induce photothermal therapy. Were the gold nanoparticles released from the prepared materials – did you test it?

16.  You state that “The ((CTS-g-GMA)-AuNPs)-Col nanocomposite at 0.6 mM was considered the best material to induce healing wounds in less time”(441-442) but it is true only for the wound up to day three. Further time-points show its performance is worse: at day 8 materials 0.3 col, 1.07 and 1.07 col outperform it; at day 18 materials 1.07 and 1.07 col induce better healing. Similar exaggeration can be found in the conclusion: “Chitosan-based Au nanocomposites presented here (with and without type I collagen) exhibit a superior ability for promoting skin would healing when compared to the positive controls TheraForm® and CTS-g-GMA”(466-468) which is true but only to some extent – please compare the values for 0.3 and 0.15.

17.  Were the studies approved by the ethics committee? There is no mention of that in the manuscript.

Other, less significant remarks

18.  The gold nanoparticles may overlap with the absorption maxima measured in the nitric oxide production assay. Was such interference evaluated?

19.  Not only “The nanocomposites (CTS-g-GMA)-AuNPs were characterized”(119) with FTIR but all the nanocomposites as I understand from the results section. Please correct.

20.  Please add GMA and collagen FTIR spectra for comparison in figure 2.

21.  Please check the paragraph in lines 242-244 as the two sentences are contradictive to each other (clustering-no clustering).

22.  Figure 6 – these are not spectra but diffractograms.

23.  When the pure AuNPs were synthesized for UV-Vis analysis, what was the reducing agent used? Please provide some experimental details (reference or description).

24.  Please check whether the meaning of the captions for figure 8 and 9 are what you were aiming to write. Currently it says that solely the AuNPs were tested. The same goes to Table 2 and 3 captions.

25.  Please explain the asterisks in table 2 footer.

26.  Please pay attention to the CREDIT statements. Currently you state that one of the authors (Moises A. Franco-Molina) did not contribute to this study, as well as nobody performed investigation, no-one prepared the original draft of the manuscript, there are no visualizations and no funding was acquired.

27.  Please delete the sentence in the line 571.

28.  English/style/typos: please check the subscripts in chemical formulas (lines 73, 99, 108, 144, 146, 169, 403); please correct either one of those – “functionalized with chitosan” (75-76) and “functionalization of CTS with GMA” (80), as they are contradictory; “nanostructured” (73-74); “hydroxides groups” (77); percent signs should be written directly after the number without the space (97); PBS – the abbreviation is explained twice, one of which with an error (137, 173); use multiplication sign (×) and not the letter x (144, 183); there are some weird signs in the pdf version of the manuscript (similar to @) – please check those (lines 199, 179); “NH3” – ammonia? (208); “y” (table 1, 386, 386); please correct the description of the y axis in figure 3; Caption of figure 8 mentions 0.6 concentration twice – should it be 0.3 and 0.6? (289); “others materials” (317); “Additionally, was found” (320-321).

Author Response

(The authors gave the same response as above.)

Reviewer 4 Report

In this study, the authors prepared CTS-g-GMA/AuNP and CTS-g-GMA-AuNP-Col nanocomposites containing a series of different HAuCl4 concentrations, and their skin wound healing capacity was studied. Cellular viability, cytokines production, and wound healing test show that the ((CTS-g-GMA)-AuNPs)-Col nanocomposite at 0.6 mM has a better skin wound healing effect in less time. In addition, the structural properties of such nanocomposites were associated with their efficacy in promoting a healing process. And it is better if the authors consider the following mentioned remarks and further improve the manuscript before submitting the final version.

1. English needs to be polished.

2. The design of the "Introduction" still needs to be improved, it is necessary to emphasize the innovative aspects of the research.

3. In the "conclusion", how to understand "these observations are traceable to the nanocomposite's electrical behavior according to electrical percolation phenomena".

4. The resolution of pictures should be adjusted.

Author Response

(The authors gave the same response as above.)

Round 2

Reviewer 3 Report

Evaluation of the response to each comment was added after each point.

The manuscript Chistosan-g-glycidyl methacrylate/Au nanocomposites promotes accelerated skin wound healing by Héctor A. López-Muñoz et al. describes the preparation of a series of hybrid composites that could be potentially utilized as wound-dressing materials. The materials were prepared and their physicochemical properties evaluated by means of XRPD, TEM, SEM FTIR and electrical conductivity measurements. The biological effects exerted by the materials included first in vitro assessment of their cytotoxicity and induction/suppression of inflammatory factors, and next by applying materials in vivo to the wounds on the backs of the balb/c mice.

Such topic is important as wound healing process using both conventional and modern means sometimes fail and possible improvements are sought for. The manuscript roughly fits the scope of the journal Pharmaceutics. It is written rather well.

However, there are many drawbacks of the study that would have to be addressed before the article can be considered for publishing:

1.- The nanocomposites preparation section should be extensively rewritten. Currently, it is impossible to recreate the experimental conditions as there is data missing – for instance the preparation of CTS-AuNPs in lines 96-101 does indeed contain the concentrations of the solution but no their volumes or ratios in which they were mixed making it pointless.

Answer: Thank you for your comment, the volumes of solutions were specified on text.

[second review] Thank you for the changes made. However, please pay attention to whether suspension or solution is reported.

2.- Please rewrite the fragment 102-106 as it unclear preparation of what is described – CTS-g-GMA? Or is CTS-g-GMA further modified? Also – what was the purpose of potassium acetate in this procedure?

Answer: We rewrote the fragment and clarified the modification of CTS-g-GMA. This reaction has been reported by the research group before, the KOH solution is used as a catalytic agent of the reaction. The OH groups are donated by potassium hydroxide opening the epoxy ring of GMA, thus producing a substitution reaction with the meridional OH of chitosan.  The following references from our group are relevant to CTS-g-GMA reaction:

Zarate Triviño, D.G.; Pool, H.; Vergara Castañeda, H.; Elizalde Peña, E. A.; Vallejo Becerra, V.; Villaseñor, F.; Prokhorov, E.; Gough, J.; García-Gaitán, B.; Luna-Barcenas, G. (chitosan-g-glycidyl methacrylate)- collagen II scaffold for cartilage regeneration. Inter. J. polym. Mat. Polym biomat. 2019, 1043-1053.

Elizalde-Peña, E. A.; Zarate-Triviño, D. G.; Nuño-DonluS. M.; Medina-Torres, L.; Gough, J. E.; Sanchez, I. C.; Villaseñor, F.; Luna-Barcenas, G. Synthesis and Characterization of a Hybrid (Chitosan-g-Glycidyl Methacrylate)–Xanthan Hydrogel. J. Biomater. Sci. Polym. Ed. 2013, 24, 1426–1442.

Flores-Ramırez, N.; Elizalde-Peña, E. A.; Vasquez-García, S. R.; Gonzalez-Hernandez, J.; Martinez-Ruvalcaba, A.; Sanchez, I. C.; Luna-Bárcenas, G.; Gupta, R. B. Characterization and Degradation of Functionalized Chitosan with Glycidyl Methacrylate. J. Biomater. Sci. Polym. Ed. 2005, 16, 473–488.

Elizalde-Peña, E. A.; Flores-Ramirez, N.; Luna-Bárcenas, G.; Vásquez-García, S. R.; Arambula-Villa, G.; García-Gaitán, B.; Rutiaga-Quinones, J. G.; González-Hernández, J. Synthesis and Characterization of Chitosan-g-Glycidyl Methacrylate with Methyl Methacrylate. Eur. Polym. J. 2007, 43, 3963–3969.3.

[second review] Thank you for the clarification. I am just wondering – the solution in which chitosan is dissolved is 0.4 M acetic acid, so an addition of KOH would result in fast reaction between acid and base forming potassium acetate. Also, the “M” is generally recognized as mol/dm3, so I have no idea what “15% of 0.05 M” solution means and am unable to assess the resulting pH of the mixture of both reagents. In the referenced article: (http://dx.doi.org/10.1163/1568562053700174) although the KOH is added, the pH is kept at 3.8 so it is very unlikely that there is unreacted KOH present in the reaction mixture. Additionally, neither of the referenced here works contains such information about the catalytic effect of KOH.

In this manner, the synthesis procedure should be discussed.

3.- “To achieve stable conditions….” (109). What were the stable conditions? How were their monitored and established?

Answer: The paragraph was corrected accordingly.

[second review] Thank you for the changes made.

4.- How much collagen was used in relation to the (CTS-g-GMA)-AuNPs to prepare the final materials? Stating only the concentration of the collagen is insufficient. Also – how much of the collagen was found to connect to the material? Was this tested anyhow? TGA-DSC would help.

Answer: The final material was prepared using 0.3% of collagen in the (CTS-g-GMA)-AuNPs formulation. However, due to natural variation of final concentrations, we do not provide an exact value of collagen because the percent is related with the final weight of each film but it is ca. 0.2-0.3% of collagen. TGA studies were reported for (CTS-g-GMA)-collagen before by Zarate Triviño et al (2019); three events were detected, namely: 1) weight loss associated with humidity (8.62%), 2) a depolymerization (47.62%) and 3) a degradation (42.76%). In comparison with (CTS-g-GMA). The results were complemented by X-ray, and FTIR studies.

Zarate Triviño, D.G.; Pool, H.; Vergara Castañeda, H.; Elizalde Peña, E. A.; Vallejo Becerra, V.; Villaseñor, F.; Prokhorov, E.; Gough, J.; García-Gaitán, B.; Luna-Barcenas, G. (chitosan-g-glycidyl methacrylate)- collagen II scaffold for cartilage regeneration. Inter. J. polym. Mat. Polym biomat. 2019, 1043-1053.

5.- The gold nanoparticle connect is never assessed nor confirmed. The amount of gold nanoparticle should be tested and reported.

Answer: The shift on FTIR bands confirmed the chemical interaction of gold nanoparticles as probed in Figure 2. The exact amount of gold nanoparticles is difficult to measure; however, Surface Plasmon Resonance (SPR) reveals the formation of nanoparticles which are corroborated by TEM micrographs. We report the nanomolar concentration of the metallic precursor used in the solution preparation; this concentration is usually reported for this kind of chemical synthesis of Au nanoparticles.

[second review] comments 4&5: Thank you for the clarification. However, in case any other research group wanted to recreate the material, e.g. for comparison reasons, any quantitative analysis could be an indicator of the successful recreation of the composite. If such are missing, it is impossible to recreate by anyone from outside your group. As for the reference (doi: 10.1080/00914037.2019.1655749), I could not find any quantitative assessment from the TGA as well.

6.- You mentioned that the impedance of the materials was measured, but no equipment is listed in lines 130-131. Also, the citation [18] is a 218 pages book – please provide at least the chapter from which the method was derived.

Answer: We corrected reference 18 accordingly and included details of electrical impedance equipment as follows:

The DC electrical conductivity (σDC) has been calculated from impedance measurements according to the methodology described by Heilmann (2003) [18]. Dielectric measurements in the frequency range from 40 Hz to 110 MHz were carried out with Agilent Precision Impedance Analyzer 4249A (Santa Clara, CA, USA). The amplitude of the measuring signal was 100 mV.

[second review] Thank you for the changes made.

7.- The “cosmetic appearance” of the healed wounds should not be mentioned so many times (lines 33, 38, 364, 442). This is a subjective matter which cannot be objectively measured thus it does noy belong in scientific article.

Answer: The purpose of this work was to develop a material with equal or better wound recovery capabilities than a commercial material. We believe we have improved the physical appearance based upon the manual on wounds and ulcers and the OECD 404 guide (it is shown in Table 1 of the Supplementary Information). Our findings help us reaffirm this conclusion since not only accelerating wound healing is necessary but also a more natural and good appearance recovery.

[second review] Thank you for the clarification. However, the wound score is mentioned once (in line 361), which shows no difference between the new composites and the reference dressing. In this case such claim is unsupported by the findings of the study and should be deleted from the paper, unless an impartial evidence is provided.

8.- Wherever “manufacturer’s instructions” are given, please provide either a description of the method or a proper reference to either these instructions or another paper where such experimental details are given (see for example lines 137 or 164)

Answer: References were added that describe the mentioned methods.

[second review] Thank you for the changes made. There are still such problems in lines 171 and 174.

9.- “peripherial blood obtained from normal donors” (136-137). Where are still talking about the mice? This not stated in the paper. Also, please define normal donors. In this manner please also improve the 2.5 subchapter, as there is no information that the spleens were harvest postmortem. Additionally – was there one mouse per one material? It is not specified in the experimental.

Answer: The paragraph was amended accordingly. The peripheral blood was obtained from human donors, the expression “normal donors” is a usual term to define that the donors do not present any pathology. The spleens were crushed, and homogenized using PBS, the supernatant of this mixture was stored for further cytometry analysis.

[second review] Thank you for the changes made. There is still no change regarding to the information about the harvesting of spleens mentioned before, as well as the number of specimens per material.

10.- Concentrations of over 5.0 mM are mentioned (line 266) but such materials were never evaluated in any studies reported.

Answer: The concentration of 5.0 mM was used to test for decrescent electrical conductivity tendency only. Due to the lack of mechanical properties, films were not prone to proper manipulation.

[second review] Thank you for the clarification.

11.- Add error bars in figure 8.

Answer: Figure 8 was amended.

[second review] Thank you for the changes made.

12.- Table 2 and 3 – why were the 0.3 mM composites not tested?

Answer: The materials at 0.3 mM were include in Tables 2 and 3 accordingly.

[second review] Thank you for the changes made.

13.- Were the results presented in figure 10 tested for their significance? You do not discuss these results in terms of effect of collagen. You state that “nanocomposite with collagen showed better results than controls” (409), yet 0.15-col showed worse results, as well as only 0.3-col and 1.07-col showed better than the controls.

Answer: The results were corrected as suggested, and the results include the materials with collagen.

[second review] Thank you for the changes made. There is still no information about the significance of the results in Figure 10. If there is no statistically significant difference between these results (similar to significance mentioned in subchapter 3.7) than one cannot discuss it in such way.

14.- You mention the mechanical properties of the materials, yet no evaluation of those was performed (i.e. tensile strength or Young’s module). To be able to discuss these properties additional test are required

Answer: In previous works from our group, the mechanical properties of this type of material have already been studied. In the discussion a short discussion of Rerefence 12 is included.

[second review] Thank you for the clarification. However, the materials reported here are not the discussed CTS-AuNPs but (CTS-g-GMA)-AuNPs and ((CTS-g-GMA)-AuNPs)-Col, which were not tested for their mechanical properties.

15.- You mention the antibacterial effect of the materials but such properties were not tested. The untreated controls did not show any bacterial infection as well, so this conclusion is exaggerated and based on other studies. The antibacterial effect could come from the CTS. You also mention the effect of visible light, but the gold nanoparticles would have to be internalized to the bacteria and the irradiated to induce photothermal therapy. Were the gold nanoparticles released from the prepared materials – did you test it?

Answer: In lines 62-63 (previous version of our manuscript), we refer to the antimicrobial properties of CTS. Similarly, on line 427 we describe those properties of AuNPs. It is important to emphasize that the antimicrobial capacity of these materials is generally discussed. The short discussion of the effect of light on AuNPs was deleted to avoid confusion. It is also important to mention that there is no control without treatment; our control on the mice wound recovery is the commercial product, which within its properties is to avoid infection in the wound. That is why no mouse suffered bacterial growth.

[second review] Thank you for the clarification and the changes made. I am a bit confused, as the caption for Figure 1 mentions the negative control and the fragment in lines 384-389 mentions the untreated wounds. Please change or clarify.

16.- You state that “The ((CTS-g-GMA-)-AuNPs)-Col nanocomposite at 0.6 mM was considered the best material to induce healing wounds in less time” (441-442) but it is true only for the wound up to day three. Further time-points show its performance is worse; at day 8 materials 0.3 col, 0.17, and 1.07 col outperform it, at day 18 materials 1.07 and 1.07 col induce better healing. Similar exaggeration can be found in the conclusion: “Chitosan-based Au nanocomposites presented here (with and without type I collagen) exhibit a superior ability to promoting skin would healing when compared to the positive control TheraForm and CTS-g-GMA” (466-468) which is true but only to some extent – please compare the values for 0.3 and 0.15.

Answer: Our conclusion was based on the dynamics of healing process. The cell migration process is a key process of wound healing; this process occurs in early stages (1 to 3 days). We included a proper reference and an explanation in text. 

[second review] Thank you for the changes made.

17.- Were the studies approved by the ethics committee? There is no mention of that in the manuscript

Answer: We added in the text the approval of the ethics committee (lines 187-190 of amended manuscript).

[second review] Thank you for the changes made.

Other, less significant remarks

18.- The gold nanoparticles may overlap with the absorption of maxima measured in the nitric oxide production assay. Was such interference evaluated?

Answer: Usually all tests performed by UV-vis require a blank solution; in our study, we used the nitric oxide reactant and gold nanoparticles at evaluated concentrations without cells.

[second review] Thank you for the clarification.

19.-Not only “The nanocomposites (CTS-g-GMA)-AuNPs were characterized” (119) with FTIR but all the nanocomposites as I understand from the results section. Please correct.

Answer: The paragraph was corrected so that the method was clearer.

[second review] Thank you for the changes made.

20.- Please add GMA and collagen FTIR spectra for comparison in figure 2

Answer: Added references related FTIR spectra of GMA and Collagen (lines 226-227).

[second review] Thank you for the changes made.

21.- Please check the paragraph in lines 242-244 as the two sentences are contradictive to each other (clustering-no clustering)

Answer: The paragraph was corrected so that the result was clearer.

[second review] Thank you for the changes made.

22.- Figure 6 – there are not spectra but diffractograms

Answer: The suggested change was made.

[second review] Thank you for the changes made.

23.- When the pure AuNPs were synthesized for UV-Vis analysis, what was the reducing agent used? Please provide some experimental details (reference or description).

Answer: They are not “pure” AuNPs, they were synthesized using CTS as a reducing agent. The paragraph was corrected.

24.- Please check whether the meaning of the captions for figure 8 and 9 are what you were aiming to write. Currently it says that solely the AuNPs were tested. The same goes to Table 2 and 3

Answer: Changes were made to the captions of Fand Tables.

[second review] comments 23&24: Thank you for the changes made in line 236 and the clarification. In such case the manuscript should be revised to reflect on the fact that no free AuNPs were studied – please revise again the captions for Figures 8 and 9.

25.- Please explain the asterisk in table 2 footer

Answer: The footer was added.

[second review] Thank you for the changes.

26.- Please pay attention to the CREDIT statements. Currently you state that one of the authors (Moises A, Franco-Molina) did not contribute to this study as well as nobody performed investigation, no-one prepared the original draft of the manuscript, there are no visualization and no funding was acquired.

Answer: Dr Moises A, Franco-Molina contribute to this study, performed investigation, prepared the original draft of the manuscript and visualization.

[second review] Thank you for the clarification. My point was to pay more attention to the CREDIT statements. Currently it seems that only Dr Moises A, Franco-Molina performed any investigation in the study, as well as prepared the original draft and was responsible for visualization. Please rewrite this section.

27.- Please delete the sentence in the line 571.

Answer: The phrase was deleted.

[second review] Thank you for the changes made.

28.- English/style/typos: please check the subscripts in chemical formulas (lines 73, 99, 108, 144, 146, 169, 403); please correct either one of these – “functionalized with chitosan” (75-76) and “functionalization of CTS with GMA” (80), as they are contradictory; “nanostructured” (73-74); “hydroxides group” (77); percent signs should be written directly after the number without the space (97); PBS – the abbreviation is explained twice, one of which with an error (137, 173); use multiplication sign (×) and no the letter x (144, 183); there are some weirds signs in the pdf version of the manuscript (similar to @) – please check these (lines 199, 179); “NH3” – ammonia? (208); “y” (Table 1, 386, 386); please correct the description of the y axis in figure 3; Caption of figure 8 mentions 0.6 concentration twice – should it be 0.6 and 0.6? (289); “others materials” (317); “Additionally, was found” (320-321)

Answer: Suggested changes were made.

[second review] Thank you for the changes made.

There are still some comments that would have to be answeared to improve the quality of the paper, as well some additional experiments would have to be performed for the study to be regarded for publishing in such a high-ranked journal as Pharmaceutics.